# Size–Pore-Dependent Methanol Sequestration from Water–Methanol Mixtures by an Embedded Graphene Slit

**DOI:** 10.3390/molecules28093697

**Published:** 2023-04-25

**Authors:** Roger Bellido-Peralta, Fabio Leoni, Carles Calero, Giancarlo Franzese

**Affiliations:** 1Secció de Física Estadística i Interdisciplinària, Departament de Física de la Matèria Condensada, Universitat de Barcelona, Martí i Franquès 1, 08028 Barcelona, Spain; 2Department of Physics, Sapienza University of Rome, Piazzale Aldo Moro 5, 00185 Rome, Italy; 3Institut de Nanociència i Nanotecnologia, Universitat de Barcelona, 08028 Barcelona, Spain

**Keywords:** Molecular Dynamics, nanoconfinement, graphene, water, methanol, sequestration

## Abstract

The separation of liquid mixture components is relevant to many applications—ranging from water purification to biofuel production—and is a growing concern related to the UN Sustainable Development Goals (SDGs), such as “Clean water and Sanitation” and “Affordable and clean energy”. One promising technique is using graphene slit-pores as filters, or sponges, because the confinement potentially affects the properties of the mixture components in different ways, favoring their separation. However, no systematic study has shown how the size of a pore changes the thermodynamics of the surrounding mixture. Here, we focus on water–methanol mixtures and explore, using Molecular Dynamics simulations, the effects of a graphene pore, with size ranging from 6.5 to 13 Å, for three compositions: pure water, 90%–10%, and 75%–25% water–methanol. We show that tuning the pore size can change the mixture pressure, density and composition in bulk due to the size-dependent methanol sequestration within the pore. Our results can help in optimizing the graphene pore size for filtering applications.

## 1. Introduction

According to the latest UN progress report on accomplishing the Sustainable Development Goals (SDGs), millions still need ‘clean water and sanitation’ (SDG 6) as well as ‘affordable and clean energy’ (SDG 7). As a result, meeting the targets by the 2030 deadline is currently unreachable. Therefore, there is an urgent need for more research and investment from governments and businesses to accelerate the implementation of these goals. SDG 6 and SDG 7 are particularly relevant for addressing some of the most pressing challenges of our time. SDG 7 seeks to ensure access to affordable, reliable, sustainable, and modern energy. This is essential for reducing greenhouse gas emissions [1], such as the byproducts of the oil industry [2], improving health, enhancing economic productivity, and creating opportunities for innovation and social inclusion. SDG 6 aims to ensure the availability and sustainable management of water and sanitation. This is vital for preventing diseases, improving hygiene, reducing inequalities, protecting ecosystems, and supporting human dignity. Furthermore, they are directly related to SDG 3 (Good health and well-being), SDG 10 (Reduced inequalities), SDG 11 (Sustainable cities and communities), SDG 12 (Responsible consumption and production), SDG 13 (Climate action), SDG 14 (Life below water), SDG 15 (Life on land), and indirectly to all the others.

Finding new and efficient ways to separate water from methanol is relevant in this context. Indeed, water and methanol are often mixed in various industrial processes, such as biodiesel production, wastewater treatment, and solvent extraction. However, separating water and methanol is challenging and energy-intensive, as they form an azeotropic mixture that cannot be easily distilled. By developing more efficient and cost-effective separation methods, such as membrane technology, adsorption, or extraction, these processes’ energy consumption and environmental impact can be reduced, contributing to the goal of affordable and clean energy. Furthermore, by recovering water and methanol from these mixtures, the quality and quantity of water resources can be improved, as well as the availability of methanol as a renewable fuel or chemical feedstock, contributing to the goal of clean water and sanitation.

Water and methanol are fully miscible liquids at ambient conditions due to their hydrogen bonds [3], and their mixtures are standard in food processing, preservation, pharmaceutical, and chemical industries. For example, water–methanol blends are used in power generation applications, including gas turbines, fuel cells, green alternative fuels, improved combustion engines, and solar plants [4,5,6,7,8,9]. Further, methanol is often added to water to lower its freezing point and improve its flow [10,11]. Nevertheless, it is necessary to separate the two components in several applications. For example, separating water and methanol is essential in producing biofuels to ensure the quality and efficiency of the final product [12,13], or in chemical manufacturing processes to maintain the desired concentration of the reactants and prevent unwanted side reactions [14].

However, the separation of methanol from water is usually performed by inefficient and energetically-intensive distillation [15]. Therefore, to minimize energy waste and increase efficiency, researchers have investigated water purification via nanomembrane filtering using chemical functionalization [16,17], adsorption on graphite pores [18], or infinite graphite sheets [19]. In general, nanomaterials and nanomembranes combined with advanced catalytic, photothermal, adsorption, and filtration processes provide fast, efficient, and tunable alternatives compared to conventional routes in water remediation. However, many challenges regarding scalability and sustainability are still open [20].

Recently, the use of graphene-based membranes has been reported [21] as a method that changes the dynamics of confined fluids when compared to the bulk. Furthermore, several works have been published exploring all types of materials, looking for the selectivity of water or methanol over the other component of the mixture [22,23,24,25]. Further, Molecular Dynamics (MD) simulations of atomistic models clarified the physical mechanisms of these changes [26] and the differences among various typologies of fluids [27,28,29]. Table 1 summarizes some of the recently obtained main experimental and theoretical results.

In particular, water has peculiar properties that are anomalous compared to other fluids [30,31,32], and its interaction with nanointerfaces dramatically modifies its structure [33], thermodynamics and dynamics [26], leading to unusual transport properties for both water and solutes [34].

On the other hand, methanol, the smallest alcohol, has an apolar methyl group (CH${}_{3}$) and a polar hydroxyl group (OH). The polar moiety can form hydrogen bonds of strength and length similar to water which, together with methanol’s small size, allow it to fully integrate into the water’s hydrogen bond network [35].

Previous works have shown that, under slit-pore confinement, water’s thermal diffusion coefficient $D_{∥}$ parallel to the walls is non-monotonic when the pore width δ changes below 1.5 nm [26]. This property is a consequence of water’s ability to form hydrogen bonds. However, recent atomistic simulations show that this behavior is also present in liquids without hydrogen bonds, including simple van der Waals liquids or not-network-forming anomalous liquids [27]. Nevertheless, the study shows that the mechanism leading to the variation of $D_{∥}$ in confined water is unique and different from other liquids [27]. Therefore, using nano-confining graphene slit-pores to separate water from methanol based on physical processes is an appealing possibility.

However, no systematic study has shown how the size of a graphene slit-pore changes the composition and thermodynamics of the surrounding mixture in which it is embedded as a solute and how to optimize it for filtering applications. This geometry reminds of the recent application of nanoengineered graphene pores used as a *sponge* for overcoming the limitations of the existing water treatment systems [20,36].

**Table 1 molecules-28-03697-t001:** Summary of experimental and simulation results with different membranes for water–methanol mixtures. List of acronyms: SA (sodium alginate), PVA (polyvinyl alcohol), PHB (poly(3-hydroxybutyrate)), rGO (reduced Graphene Oxide), CS (chitosan), ZIF-n (Zeolitic Imidazolate Framework, where the number “n” is not related to the structure, just used as naming [37]), PVDF (polyvinylidene fluoride), BNNS (boron nitride nanosheets, functionalized with -H,-F groups and -OH group), BNNT (boron nitride nanotubes). Symbols ↑, ↓, ⇒ stands for *increasing, decreasing, implying*, respectively.

Material	Method	Selective to	Main Result	Ref.
Porous BNNS -H, -F -OH	Simulations	Methanol Water	Each molecule has higher free energy in correspondence to the pores it cannot permeate through.	[16]
BNNT	Simulations	Alcohols	Alcohols can easily break their hydrogen bonds to enter and occupy the nanotubes, having a strong interaction with them.	[17]
Pristine graphene	Simulations	Methanol	Methanol gets preferentially absorbed into a graphene slit pore. When mixed with water, the two liquids couple and diffuse.	[18]
Graphite plates	Simulations	Methanol	Preferential absorption of methanol on graphite sheets due to Van der Walls interactions between the methyl groups and the carbon.	[19]
GO	Experiments	Water	Low friction flow of a water monolayer through 2D channels between graphene sheets, while helium remains in feed.	[21]
SA/PVA	Experiments	Water	↑ T ⇒ ↑ mobility of the polymer chain⇒ ↑ flux, little selectivity reduction. At 5% PVA composition, the material has surface pores, and at 20% has cracks. Optimum PVA composition at 10%.	[22]
PHB	Experiments	Water	Pure substance pervaporation shows good MeOH permeation. It has water selectivity in a mixture due to the hydrogen bond network. MeOH has reduced mobility when mixed with water.	[23]
rGO/CS	Experiments	Water	The interlayer space due to the CS leads to molecularly sieve water, and the hydrophobicity of GO provides good flux.	[24]
ZIF-8/PVDF ZIF-67/PVDF	Experiments	Water	ZIF-67/PVDF membrane enhances flux due to its hydrophilicity. However, ↑ water % in the feed ⇒ ↑ swelling ⇒ ↓ selectivity as volume increases and MeOH molecules can also pass through. ↑ T ⇒ ↑ polymer chain mobility ⇒ ↑ flux, ↓ selectivity.	[25]

Here, we investigate the capacity of a graphene slit-pore to sequester methanol from a water–methanol mixture and its effects on the mixture. In particular, we study how different water–methanol mixture properties—such as density, pressure, and composition—are affected as the width δ of an embedded graphene slit-pore changes. To this end, we perform Molecular Dynamics (MD) simulations of water–methanol mixtures of different compositions with graphene slit-pores of different sizes as solutes. The water–methanol mixture is described using tested coarse-grained models (see Section 3) based on Continuous Shouldered Well (CSW) and Lennard–Jones potentials.

## 2. Results and Discussion

### 2.1. Number Density

#### 2.1.1. The Pure CSW Case

First, we check the behavior of the pure CSW (representing pure water-like fluid, see Section 3) in the subregion $V^{\prime }$ outside the pore (Figure 1a) and find that the slit-pore width δ affects weakly its density (Figure 1a,b). In particular, the CSW density in $V^{\prime }$ has a minor decrease for increasing δ, but remains close to the overall density of 0.036 ${Å}^{-3}$ within the error, which is the nominal number density in the entire simulation box, including the slit-pore (see Section 3). Therefore, for the considered range of δ, the nano-pore does not adsorb much water-like liquid inside, consistent with its hydrophobic properties at the macroscopic scale.

Nevertheless, in Ref. [27] a similar confined liquid (CSW with Δ=30) shows free energy and confined density extrema at ∼7.5 Å, ∼9.5 Å, ∼10.5 Å, and ∼12 Å that seems to correlate with the small density variations we find here, although within the error bars. Hence, we explore, next, the behavior of the mixture in $V^{\prime }$ to better investigate the pore-size effects.

#### 2.1.2. The Mixture Case

At some fixed pore sizes, e.g., δ=9 Å, we find an overall decrease for the mixture density $\rho_{V^{\prime }}$ when we increase the methanol concentration from 0% to 10% and 25% (Figure 1a). This behavior is consistent with what is expected for bulk [38,39]. At other values of δ, e.g., δ=7 Å, the trend is nonmonotonic in methanol concentration, suggesting an unexpected behavior.

Indeed, surprisingly, we observe that changes in the slit-pore width δ affect the mixture $\rho_{V^{\prime }}$ outside the error bars. Moreover, the effect in the mixture is more evident than in pure CSW liquid. Therefore, the pore-size dependence of the density in $V^{\prime }$ depends mainly on the methanol-pore interaction.

To better understand this dependence, we calculate the concentration of each component in $V^{\prime }$ separately as a function of δ. The overall CSW number density in the subvolume $V^{\prime }$, $\rho_{CSW}$, should decrease from 0.036 ${Å}^{-3}$ at 0% methanol concentration, to 0.0324 ${Å}^{-3}$ at 10% methanol, to 0.027 ${Å}^{-3}$ at 25% methanol. However, we find that at the highest methanol concentration, $\rho_{CSW}$ is ≈4% above the expected value (Figure 1b). Therefore, there is less methanol in the solution than expected due to the presence of the graphene pore.

Furthermore, $\rho_{CSW}$, at a nominal 25% methanol concentration, tends to increase for larger pore sizes δ (Figure 1b). Hence, the amount of methanol in the mixture in $V^{\prime }$ decreases for greater δ, and the strength of the effect is proportional to the nominal methanol concentration.

This conclusion is confirmed when we explicitly calculate the methanol number density in $V^{\prime }$, $\rho_{meth}$, (Figure 1c). The expected values for $\rho_{meth}$ are 0.0036 ${Å}^{-3}$ at 10% methanol concentration and 0.0090 ${Å}^{-3}$ at 25% methanol. However, we find that $\rho_{meth}$ can be as low as ≈83% of the expected values.

In particular, the decrease of $\rho_{meth}$ in both cases is non-monotonic, with maxima and minima that partially correlate with the confined CSW density extrema at ∼7.5 Å, ∼9.5 Å, ∼10.5 Å, and ∼12 Å found in Ref. [27], suggesting that the polar part of the methanol interaction could partially be responsible for this non-monotonicity. Furthermore, the decrease is more evident when the nominal methanol concentration is higher, in agreement with the $\rho_{CSW}$ behavior. Finally, the overall $\rho_{meth}$ decreases for increasing δ, suggesting an increased absorption of methanol inside the graphene pore when its size increases.

To test this result directly, we calculate the methanol sequestered by the interior walls of the pore as a function of δ for the two mixture compositions (Figure 2, left panel). We find that methanol in the pore exceeds what would be expected from simple osmotic equilibrium. In particular, at the overall 10% and 25% methanol compositions, we find up to 320% and 250% more methanol than expected inside the pore, respectively. Furthermore, the greater δ, the higher the methanol sequestered, above the value expected by simple osmotic diffusion. Hence, the pore inside adsorbs more methanol when its size increases.

Consequently, the methanol amounts inside and outside the pore are anti-correlated. The effect in $V^{\prime }$ is more substantial for higher methanol overall concentration (Figure 2). For increasing slit size, the trend for the methanol concentration is to increase inside and decrease outside the pore. However, the methanol concentration changes non-monotonically, correlating with the oscillatory properties of the confined CSW liquid [27].

### 2.2. Pressure in $V^{\prime }$

Our results show a possible correlation between the observed variation of the mixture density in $V^{\prime }$ and the properties of the confined CSW model adopted for the polar interactions of the mixture components. Although the CSW model has been tested in the literature for the mixture bulk-properties [40], there is no study about its reliability when the bulk embeds a graphene pore. Therefore, we calculate the mixture’s pressure $P_{V^{\prime }}$ in the subvolume $V^{\prime }$ to test if the coarse-grained model qualitatively reproduces the correct thermodynamics (Figure 3).

First, we find that $P_{V^{\prime }}$ increases as we increase the amount of methanol in the system. This is consistent with the expected thermodynamics [38,39].

At 0% and low, 10%, methanol concentration, the overall $P_{V^{\prime }}$ does not change significantly within the error bars when the slit-pore width varies. However, for 25% methanol concentration, the $P_{V^{\prime }}$ tends to decrease weakly for increasing pore’s width δ.

Because our calculations show that the density of the mixture in $V^{\prime }$ weakly oscillates with δ (Figure 1a), to test the mixture’s equation of state, at least qualitatively, we make a parametric plot of $P_{V^{\prime }}$ as a function of $\rho_{V^{\prime }}$ (Figure 4). Within the range of the observed δ-dependent variations, we find a behavior that is qualitatively consistent with the existing experimental results, being $P_{V^{\prime }}$ an increasing function of $\rho_{V^{\prime }}$ that is approximately linear within the error bars [38,39].

Therefore, the coarse-grained model preserves the mixture equation of state qualitatively under the thermodynamic conditions we explore here. This occurs despite the lack of directionality in the polar interaction. As discussed in Ref. [41], a consequence of this approximation is the incorrect estimate of the polar entropic contribution to the free energy of the mixture, as demonstrated in Ref. [27] when comparing the free energy of the confined CSW potential with that of the atomistic TIP4P/2005 water model.

## 3. Materials and Methods

In this study, we use a coarse-grained model for the water–methanol mixture [40] in which both molecules are represented schematically as beads, one for water and two for methanol. Here, the OH is modeled using the Continuous Shouldered Well (CSW) potential [42], which has been used to reproduce several properties of systems as different as liquid metals, colloids, or, as in our case of interest, water (and hydroxyl functional groups) [35,43].

Although it does not reproduce all the water properties due to its lack of directionality [41], the CSW represents a simple approximation that significantly reduces the computation time in MD simulations compared to atomistic models. Furthermore, it is simple enough to be studied analytically, as shown, for example, in Refs. [43,44,45,46].

### 3.1. The Coarse-Grained Models for the Mixture

Following previous studies of water–methanol mixtures [40], we represent a water molecule as a single (polar) CSW bead, while methanol as a dumbbell (two touching beads) made of an apolar 24-6 Lennard-Jones (LJ) bead for the CH${}_{3}$ moiety and a polar CSW bead for the OH group. The CSW potential for the polar-polar (OH-[OH, H${}_{2}$O], H${}_{2}$O-H${}_{2}$O) interactions is defined as (Figure 5, red line)
(1)$$U_{\mathrm{CSW}}\left(r\right)\equiv \frac{U_{R}}{1+\exp\left(\frac{\Delta (r-R_{R})}{a}\right)}-U_{A}\exp\left(-\frac{{\left(r-R_{A}\right)}^{2}}{2\omega_{A}^{2}}\right)+U_{A}{\left(\frac{a}{r}\right)}^{24},$$
with parameters
$$\begin{array}{cc} \frac{U_{R}}{U_{A}} & =2,\;U_{A}=0.2\;\mathrm{kcal}/\mathrm{mol},\\ \frac{R_{R}}{a} & =1.6,\;a=1.77\;Å,\\ \frac{R_{A}}{a} & =2,\;{\left(\frac{\omega_{A}}{a}\right)}^{2}=0.1,\\ \Delta & =15, \end{array}$$
where *a* stands for the hard-core distance (the diameter of the particles); $R_{R}$ and $R_{A}$ are the repulsive radius and the distance of the attractive minimum, respectively; $U_{R}$ and $U_{A}$ are the energy of the repulsive shoulder and the attractive well, respectively; Δ controls the softness of the potential at $R_{R}$; and $\omega_{A}^{2}$ is the variance of the Gaussian centered in $R_{A}$ [42,47]. The values of the parameters for the CSW model are set in agreement with previous works [27,35,40,46]. Specifically, $U_{A}$ and *a* are chosen to allow a comparison with atomistic water models [27], and Δ is as in Refs. [35,46] to benchmark our results against those for bulk.

The CH${}_{3}$-CH${}_{3}$ interaction (Figure 5, black line) is described by the 24-6 LJ potential [35,40,46]
(2)$$U_{\mathrm{LJ}}\left(r\right)\equiv \frac{4}{3}2^{2/3}\epsilon_{\mathrm{LJ}}\left[{\left(\frac{\sigma_{\mathrm{LJ}}}{r}\right)}^{24}-{\left(\frac{\sigma_{\mathrm{LJ}}}{r}\right)}^{6}\right],$$
with
$$\begin{array}{cc} \frac{\sigma_{\mathrm{LJ}}}{a} & =1.0,\;\sigma_{\mathrm{LJ}}=1.77\;Å,\\ \frac{\epsilon_{\mathrm{LJ}}}{U_{A}} & =0.1,\;\epsilon_{\mathrm{LJ}}=0.02\;\mathrm{kcal}/\mathrm{mol}, \end{array}$$
where the values of $\sigma_{LJ}$ and $\epsilon_{LJ}$ are chosen such to compare with the CSW parameters.

The CH${}_{3}$-[OH, H${}_{2}$O] interaction is modeled with the 24-6 LJ potential employing the Lorentz–Berthelot mixing rules (Figure 5, green line)
(3)$$U_{mix}\left(r\right)\equiv \frac{4}{3}2^{2/3}\epsilon_{mix}\left[{\left(\frac{\sigma_{mix}}{r}\right)}^{24}-{\left(\frac{\sigma_{mix}}{r}\right)}^{6}\right],$$
with
$$\begin{array}{cc} \sigma_{mix} & \equiv \frac{1}{2}\left(\sigma_{\mathrm{LJ}}+a\right)=1.77\;Å,\\ \epsilon_{mix} & \equiv \sqrt{\epsilon_{\mathrm{LJ}}U_{A}}=0.06\;\mathrm{kcal}/\mathrm{mol}. \end{array}$$

We consider three mixture compositions, 100% CSW, 90%–10% CSW-methanol, and 75%–25% CSW-methanol. The first case allows us to establish a benchmark, while the other two can be compared against the bulk cases in Ref. [40]. Regarding the relevance in actual cases, the 90%–10% composition can be compared to a mildly polluted water mixture. On the other hand, the more concentrated mixture with 75%–25% composition is possibly relevant in industry processes.

### 3.2. The Model for the Graphene Slit Pore

Each graphene sheet is modeled as a honeycomb lattice in agreement with its atomic structure. Each graphene atom interacts with the fluid particles via the standard 12–6 LJ potential (Figure 6, violet line)
(4)$$U_{\mathrm{LJ}}^{\mathrm{graphene}}\left(r\right)\equiv 4\epsilon_{g}\left[{\left(\frac{\sigma_{g}}{r}\right)}^{12}-{\left(\frac{\sigma_{g}}{r}\right)}^{6}\right],$$
with $\sigma_{g}=3.26\;Å$ and $\epsilon_{g}=0.1$ kcal/mol [27], as established in literature [48]. The positions of graphene atoms are kept fixed during the simulation.

### 3.3. Graphene Slit-Pore Geometry

The graphene slit-pore is composed of two parallel sheets of sizes $l_{x,gr}=49\;Å$, $l_{y,gr}=51\;Å$ and width $l_{z,gr}=\delta$. We vary δ from 6.5Å up to 13Å. The pore is included in a volume $V=L_{x}\;L_{y}\;L_{z}$, with $L_{x}=L_{y}=84$ Å and $L_{z}=98$ Å and periodic boundary conditions (p.b.c.).

All the observables are calculated in a subvolume $V^{\prime }\equiv L_{x}^{\prime }\;L_{y}^{\prime }\;L_{z}^{\prime }$ sufficiently separated from the graphene walls to avoid direct interface effects, with $L_{x}^{\prime }=L_{y}^{\prime }=84$ Å, and $L_{z}^{\prime }=44\;Å$ (Figure 7). With this choice of parameters, the distance of $V^{\prime }$ from a graphene wall is always >20 Å.

To evaluate the changes of properties in $V^{\prime }$, we compute the number density of the mixture and of both of its components as:(5)$$\rho_{\alpha }\equiv \frac{\langle N_{\alpha }\rangle }{V^{\prime }},$$
where $\langle N_{\alpha }\rangle$ is the ensemble average of the number of molecules α, with α standing for CSW, methanol, or both (for which we use the symbol $\rho_{V^{\prime }}$), inside the subregion $V^{\prime }$. We also calculate the pressure of the system,
(6)$$P_{V^{\prime }}=\frac{\langle N\rangle k_{B}T}{V^{\prime }}+\frac{1}{3V^{\prime }}\langle \sum_{i=1}^{N}\sum_{i>j}^{N}r_{ij}\cdot f_{ij}\rangle .$$

The first (kinetic) term comprises the ensemble average of all molecules $\langle N\rangle$, the Boltzmann constant $k_{B}$, the fixed temperature of the simulation *T*, and the volume of the subregion $V^{\prime }$. The second term is the Virial, averaged over the ensemble of all the pairs of molecules *i*, *j* in the volume $V^{\prime }$, and $r_{ij}\cdot f_{ij}$ the scalar product between their distances and forces. Finally, we compute the mixture composition in the subregion $V^{\prime }$ outside the pore (Figure 7).

For comparison, we also calculate the amount of methanol inside the pore as a function of the width δ. To minimize the edge effects of the walls, we consider only a reduced region of the slit-pore, i.e., a central subvolume $V_{s}=L_{x,s}\;L_{y,s}\;\delta$, where $L_{x,s}=L_{y,s}=30\;Å$.

### 3.4. Molecular Dynamics

We analyze the system by Molecular Dynamics (MD) simulations, where Newton’s equations of motion of each degree of freedom are solved numerically in the canonical ensemble with a fixed total number *N* = 25,000 of particles composing the mixture, in a total volume *V*, at fixed temperature T=100 K controlled by the Nosé–Hoover thermostat, as implemented in the LAMMPS software [49]. For the graphene slit-pore in the simulation box with p.b.c., we set the initial configuration by intertwining two hexagonal lattices (maximum packing) of CSW beads and methanol dimers and melting them during the initial equilibration procedure. Without the slit-pore, the number density would be N/V=0.036 Å${}^{-3}$.

We adopt the Leap–Frog integration algorithm [50], a standard second-order and time-reversible integration method, with time-step δt=1 fs. We equilibrate the system for 0.1 ns to properly mix the two liquids and analyze the data collected every 100 fs for the next 0.1 ns.

## 4. Summary and Conclusions

Water–methanol separation is relevant to several industrial applications, including methanol extraction for biofuels [4,5]. However, traditional methods have limited efficiency and high economic costs [15]. Therefore, exploring alternative approaches is technologically significant, scientifically challenging, and timely for the UN Sustainable Development Goals (SDGs) “Clean Water and Sanitation” and “Affordable and clean energy”. In particular, graphene sponges are novel nanomaterials that can effectively remove contaminants from water. They have several advantages over conventional methods, such as high surface area, tunable pore size, and multifunctional properties. Moreover, they can combine different water purification mechanisms, such as adsorption, and catalytic and electrocatalytic degradation of pollutants. This makes them promising candidates for various environmental engineering and water treatment applications [20,36].

Here, we investigate by Molecular Dynamics how an embedded graphene slit-pore modifies the properties of a water–methanol mixture. We find that the preferential interaction of the graphene with the hydrophobic moiety of the methanol induces an effective decrease of methanol concentration in the solution. On the other hand, the methanol accumulation inside the pore can be as high as 320% of its nominal concentration under the condition we explore here, leading to a methanol depletion in the solution composition up to ≃84% of its overall value. Consequently, the methanol concentrations inside and outside the pore are anti-correlated with a more evident effect for higher methanol overall concentration (Figure 2).

We observe that the slit-pore width δ has an appreciable effect on all the solution thermodynamic quantities of the mixture and its components outside the pore. In particular, their densities (Figure 1a–c) and the mixture pressure (Figure 3) decrease for increasing δ. Therefore, these quantities can be tuned by changing the size of the embedded pore due to the size-dependent adsorption of methanol within the graphene-confined region.

Because we coarse-grain the mixture based on the CSW water-like liquid [42] and the CSW-based dumbbell methanol model [35] to make the simulation efficient, we test if these results could be due to the approximation of our approach. Indeed, both models have been tested in the literature [46,51], reproducing properties of the mixture [18,38,39,40]. Nevertheless, the lack of a directional interaction, leading to the formation of a hydrogen bond network, makes these models able to reproduce only part of the properties of the two components of the mixture, as discussed for the case of water, for example, in Refs. [27,41]. We, therefore, test if the changes we find for the mixture density and pressure outside the pore as a function of the pore size are thermodynamically consistent. We find that they follow (qualitatively) the expected equation of state of the mixture, validating our results. These results encourage further investigation to find the optimal parameters for graphene slit-pore and sponge applications in nano-filtering and purification of water–alcohol mixtures by physical mechanisms. In addition, further investigation will be necessary to understand if, for example, the sequestration of methanol within the graphene layers increases for an increasing number of layers.

## Figures and Tables

**Figure 1 molecules-28-03697-f001:**
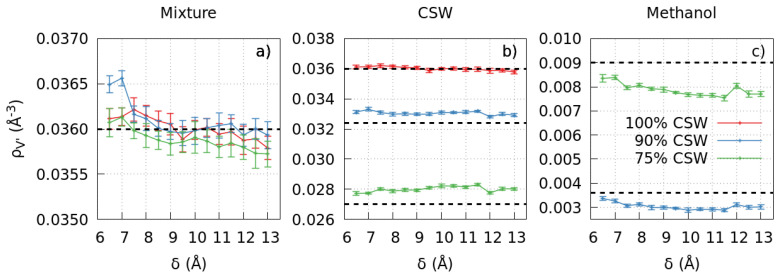
The slit-pore width affects the densities of the mixture outside the pore. For pure CSW (red) and CSW-methanol compositions 90%–10% (blue) and 75%–25% (green), the change in δ implies variations of density $\rho_{V^{\prime }}$ of the mixture (panel **a**) and each component (CSW, panel **b**, and methanol, panel **c**) outside the pore. In each panel, horizontal black dashed lines mark the values of $\rho_{V^{\prime }}$ that would be expected without the embedded slit-pore for the mixtures with different compositions as indicated in the legend in panel **c**, for the mixture (**a**), the CSW (**b**), the methanol (**c**).

**Figure 2 molecules-28-03697-f002:**
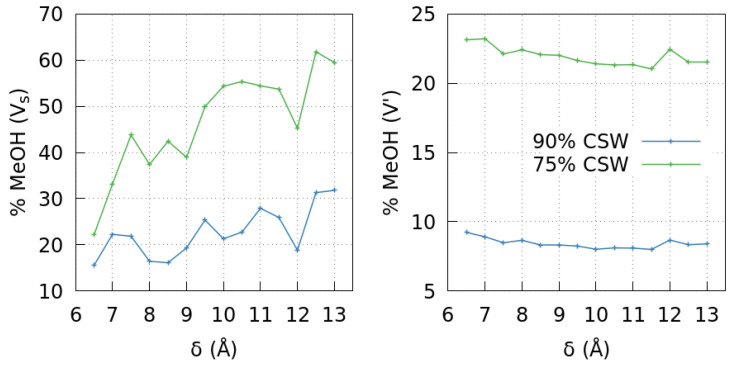
The sequestration of methanol inside the graphene slit-pore affects its concentration outside. The methanol concentrations inside (**left** panel) and outside the pore (**right** panel) are anti-correlated for both mixture compositions 90%–10% (blue) and 75%–25% (green) CSW-methanol. Inside the pore, the concentration can increase by ≃320%, in the first case, compared to its nominal bulk value and 250%, in the second case, while outside can be ≃75% or 84% lower, respectively.

**Figure 3 molecules-28-03697-f003:**
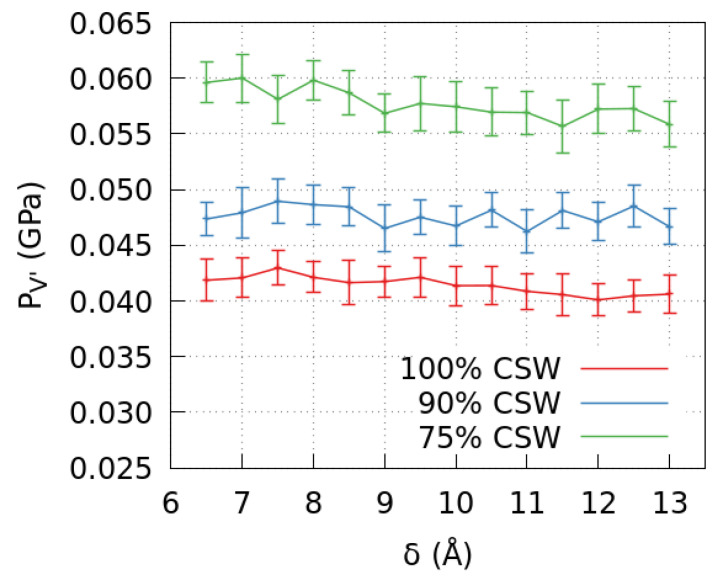
The pressure in $V^{\prime }$ at different mixture compositions and graphene slit-pore sizes. $P_{V^{\prime }}$ increases when the methanol concentration goes from 0% to 10%, to 25%, and has a weak dependence on the pore’s width δ. The dependence is more evident for higher methanol concentrations.

**Figure 4 molecules-28-03697-f004:**
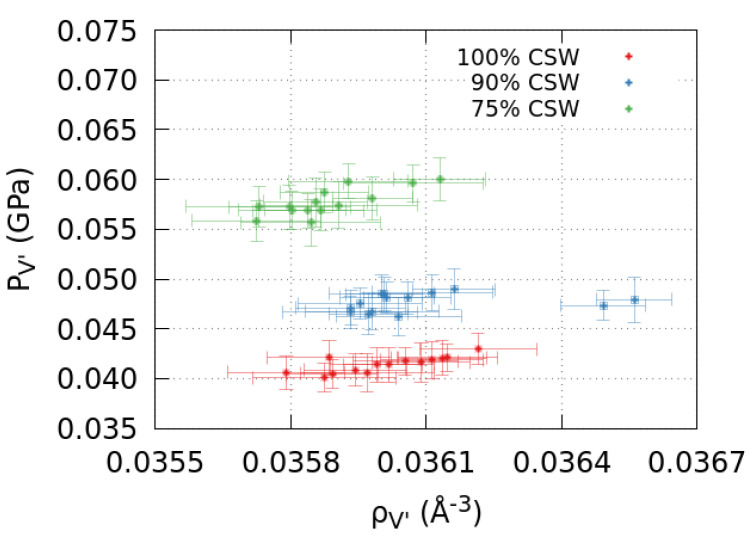
The pressure and density of the mixture in $V^{\prime }$ behave as expected. We calculate the density $\rho_{V^{\prime }}$ and pressure $P_{V^{\prime }}$ in $V^{\prime }$ for pure CSW (red) and CSW-methanol mixtures with compositions 90%–10% (blue) and 75%–25% (green) for different slit-pore widths (Figure 1a and Figure 3, respectively). The parametric plot shows that the two quantities are proportional and, at fixed $\rho_{V^{\prime }}$, $P_{V^{\prime }}$ increases for increasing methanol concentration, consistent with experiments in bulk [38,39].

**Figure 5 molecules-28-03697-f005:**
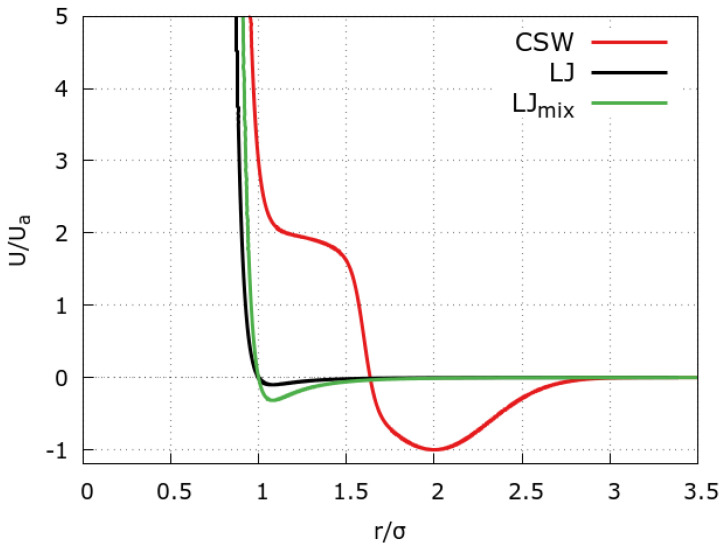
Interaction potentials between the fluid beads. The CSW potential (red line) has two characteristic length scales (the repulsive shoulder and the attractive well) as the hydrogen bond between polar groups in methanol and water. The 24–6 Lennard–Jones (LJ, black line) and 24–6 LJ with Lorentz–Berthelot (LB) mixing rules (green line) are the interaction potentials for the methyl–methyl and methyl–hydroxyl interactions, respectively.

**Figure 6 molecules-28-03697-f006:**
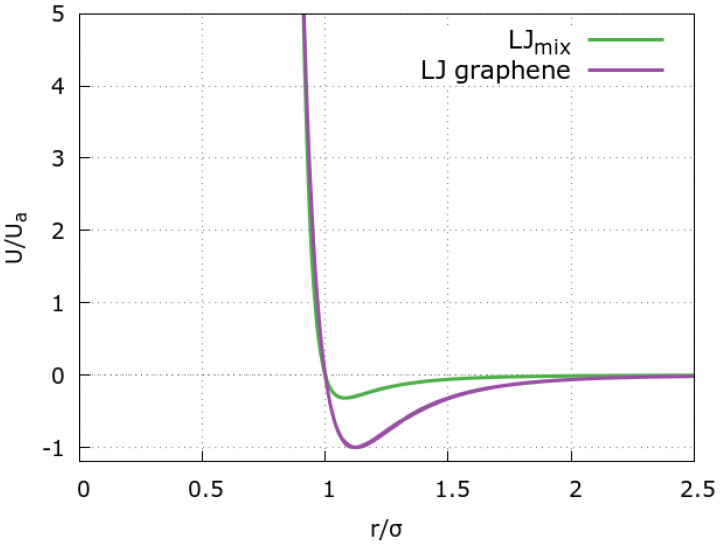
Interaction potentials between the fluid beads and the graphene atoms. The 12–6 LJ graphene-fluid interaction (violet line) is compared with the mixing potential in Figure 5 (green line).

**Figure 7 molecules-28-03697-f007:**
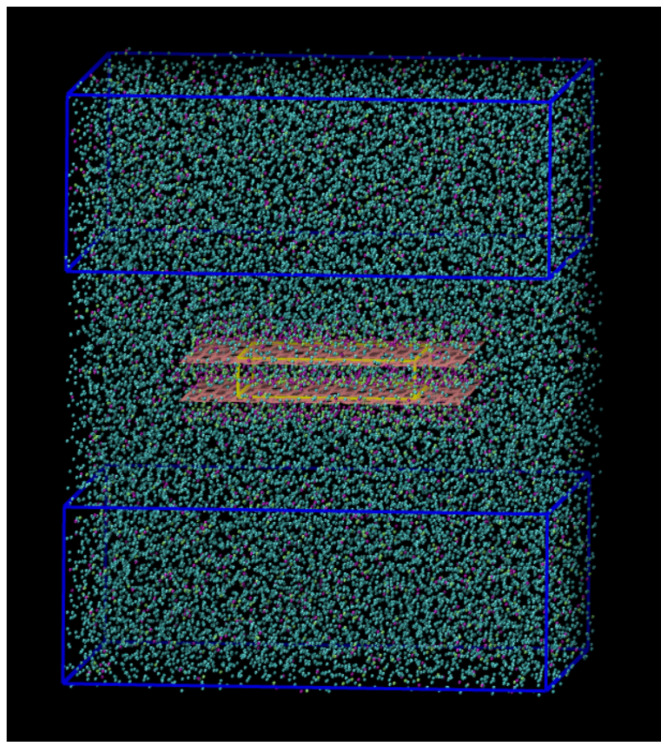
Simulation snapshot for a 90%–10% CSW-methanol mixture. The two navy blue boxes, connected by the periodic boundary conditions, correspond to the subregion with volume $V^{\prime }$ in which we calculate the observables. Blue beads are CSW particles; methyl groups are purple; hydroxyl groups are green; pink lines represent the graphene lattice. The yellow box emphasizes the subvolume $V_{s}$ in which we compute the confined mixture composition.

## Data Availability

Data supporting reported results are available upon request.

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
