# Peer review of "Size–Pore-Dependent Methanol Sequestration from Water–Methanol Mixtures by an Embedded Graphene Slit"

_molecules, 2023, doi:10.3390/molecules28093697_

Round 1

Reviewer 1 Report

This manuscript authors proposed a technique for separation water-methanol mixtures with pore sizes controllable GO. The theory adopted, the experiments and the MS simulation designed were reasonable and the results have some interest in the water-methanol separation industrial applications. I suggest the manuscript be published in this journal after some small revision.

First, I was not so understand why authors ordered the citations start from [42], this issue should be checked and revised.

Second, recommend authors to provided some more detailed experimental results to compare or fit for the calculated results.   

Third, the English language can be improved for readability.

Reviewer 3 Report

Dear authors,

Thank you for the opportunity to read your manuscript. The research is interesting and I was interested to read it, but I have a number of questions for successful publication. 

Manuscript molecules-2334221, Title "Size-pore-dependent methanol sequestration from water-methanol mixtures by an embedded graphene slit" by authors Roger Bellido Peralta, Fabio Leoni, Carles Calero, Giancarlo Franzese. 

In this paper the authors presents investigations on water-methanol mixtures and explore and also Molecular Dynamics simulations, the effects of a graphene pore, with size ranging from 6.5 to 13 Ã…, for three compositions: pure water, 90%-10%, and 75%-25% water-methanol. The authors show that tuning the pore size can change the mixture pressure, density, and composition in bulk due to the size-dependent methanol sequestration within the pore. The results of this study can help in optimizing the graphene pore size for filtering applications.

STRENGTHS:

The authors have a good understanding of the subject area. The authors have carried out interesting research. The paper is noteworthy.

WEAKSIDES:

1. The first thing that catches the eye is the ambiguity in the references to the works. Why do you start the reference with work 42? Where are 1,2,3... etc.

2. First, a systematic analysis of the literature is needed. Give specific works, specific authors, who have dealt with the issue and what conclusions they have reached. And you have a general analysis in the Introduction section. It is confusing.  It is good to present the literature analysis of the most interesting works in your opinion in the form of a table. Which author, which work, and which result.

3. The links to the pictures need to be organised. You have Figure 2 referring to sub-section 2.2, while it is listed in sub-section 2.1. Line 87-97

The same applies to figure 4.

4. I am not quite clear from the paper what is the experimental equilibrium concentration of methanol when the components are soluble?

5. At concentrations of methanol in the aqueous phase up to 30 wt. %, the hydrocarbon content in the aqueous layer is extremely low, as well as water content in the organic layer, and does not exceed 0.3 wt. %. Under these conditions the phases formed were treated as binary mixtures, and simpler equations to describe equilibrium in ternary systems.

6. Did you use the Newman (purely molecular diffusion), Cronig-Brink (laminar circulation in droplets) and Handlos-Baron (fully developed turbulence) models in your studies?

Also, to improve the reference list, I recommend that you look at a few articles that I think might improve the reference list:

Fetisov, V, Gonopolsky, AM, Davardoost, H, Ghanbari, AR, Mohammadi, AH. Regulation and impact of VOC and CO2 emissions on low-carbon energy systems resilient to climate change: a case study on an environmental issue in the oil and gas industry. Energy Sci Eng. 2023; 1- 20. doi:10.1002/ese3.1383

Fetisov, V.; Gonopolsky, A.M.; Zemenkova, M.Y.; Andrey, S.; Davardoost, H.; Mohammadi, A.H.; Riazi, M. On the Integration of CO2 Capture Technologies for an Oil Refinery. Energies 2023, 16, 865. https://doi.org/10.3390/en16020865

Round 2

Reviewer 3 Report

Dear authors, 

I am satisfied with the corrections and I think your manuscript should be published in the journal.

Best regards, reviewer